# Fast Capture and Efficient Removal of Bloom Algae Based on Improved Dielectrophoresis Process

**DOI:** 10.3390/ijerph20010832

**Published:** 2023-01-01

**Authors:** Jinxin Liu, Qinghao Jin, Junfeng Geng, Jianxin Xia, Yanhong Wu, Huiying Chen

**Affiliations:** 1Department of Environmental Science, College of Life and Environmental Science, Minzu University of China, Beijing 100081, China; 2Institute for Materials Research and Innovation, Institute for Renewable Energy and Environmental Technologies, University of Bolton, Deane Road, Bolton BL3 5AB, UK

**Keywords:** eutrophication, cyanobacteria, *Anabaena* removal, Dielectrophoresis, water purification

## Abstract

A dielectrophoresis (DEP) method for direct capture and fast removal of *Anabaena* was established in this work. The factors affecting the removal efficiency of *Anabaena* were investigated systematically, leading to optimized experimental conditions and improved DEP process equipment. The experimental results showed that our improved DEP method could directly capture *Anabaena* in eutrophic water with much enhanced removal efficiency of *Anabaena* from high-concentration algal bloom-eutrophication-simulated solution. The removal rate could increase by more than 20% after applying DEP at 15 V compared with a pure filtration process. Moreover, the removal rate could increase from 38.76% to 80.18% in optimized experimental conditions (the initial concentration of 615 μg/L, a flow rate of 0.168 L/h, an AC voltage of 15 V, and frequency of 100 kHz). Optical microscopic images showed that the structure of the captured algae cells was intact, indicating that the DEP method could avoid the secondary pollution caused by the addition of reagents and the release of phycotoxins, providing a new practical method for emergent treatment of water bloom outbreaks.

## 1. Introduction

Nutrient-rich sewage and waste are fed into closed, slow-moving water bodies such as lakes, bays, and shallow rivers, resulting in sharp depletion of oxygen due to excessive Phytoplankton growth and weed growth, and leading to a condition known as eutrophication [1]. The International Organization for Standardization (ISO) defined eutrophication as algae and other higher plants accelerating their growth in nutrient-rich water [2]. The major sources of eutrophication are the runoff of nitrogen and phosphorus fertilizers from farmland, the random discharge of untreated livestock manure, and discharge of substandard industrial wastewater.

Eutrophication and related ecological health risks have become some of the biggest challenges to sustainable water resources management [3]. The substances containing nitrogen and phosphorus exceeding the self-purification capacity in the water body are the material basis for the outbreak of water blooms [4]. Among the algae produced during water bloom, cyanobacteria are nutritionally superior to other planktonic photosynthetic species, so they dominate the population, and their chlorophyll-a content can be measured to reflect the severity of the bloom [5]. Among them, *Anabaena flos-aguas*, *Anabaena*
*spiroides* and *Anabaena circinalis* are the main algal organisms that form cyanobacteria bloom because of their strong nitrogen fixation, fixing nitrogen in the air to ammonia nitrogen [6]. So the solution of *Anabaena flos-aguas* was used as eutrophication simulated wastewater in this study.

The outbreak of cyanobacteria blooms will cause water-quality problems [7], which are persistent and serious [8]. In addition, cyanobacteria blooms produce large amounts of toxins of cyanobacteria (cyanotoxins), which can lead to liver, digestive and neonatal diseases when ingested by birds, mammals and humans [9]. Cyanotoxins are classified as hepatotoxins, cytotoxins and neurotoxins, according to the type of injury [10].

Many methods have been applied to alleviate cyanobacteria pollution, including hydro physical and physical control [11,12], such as harvesting of the floating cells [13]; chemical treatment [14,15], such as nutrient removal [16,17]; and biological treatment [18]. However, the shortcomings of these methods limit their application in the treatment of eutrophic water. The removal of plankton cells and nutrients may produce serious ecotoxicological effects and reduce the nutrient load of the lake. The high cost of the physical method makes it difficult to use in large-flow water [19]. Although the chemical method has high removal efficiency, it easily produces secondary pollution, while the biological method requires a long period and is greatly affected by natural conditions, so it may not meet the needs of sudden bloom remediation [20]. Therefore, it is of great significance to develop a low-cost and efficient method to reduce cyanobacteria pollution.

The dielectrophoresis (DEP) method refers to the directional migration of particles subjected to dielectric polarization in a non-uniform electric field [21]. We would like to simply state the following mechanism to explain the DEP process at cellular level: The non-uniform electric field generated in the DEP device polarizes the cells and induces a dipole moment on each cell. As a result, cells in the electric field are subjected to an unbalanced force exerted by the electric field, which drives them to move along the electric field gradient in the solution. Due to the strong electric field generated in the crossed area of the mesh electrodes, those cells close to the electrodes are the first to begin to migrate directionally. In addition, the polarization induction effect between adjacent cells causes the cells at a greater distance to be polarized, and subsequently migrate directionally. In this way, more and more cells are enriched in certain areas of the device, thus achieving the purpose of cell separation. The DEP force and direction received by the cell in a non-uniform electric field can be explained by the following equation [22]:(1)FDEP =2πR3εmRe[K(ω)∇Erms2]
where *R* denotes the radius of the particle, $\varepsilon_{m}$ refers to the dielectric constant of a suspended medium, $\nabla E_{\mathrm{rms}}^{2}$ is the gradient of the square of electric field strength, while Re[K(ω)] is the real component of the complex Clausius–Mossotti (CM) factor, given by:(2)$$K\left(\omega \right)=\frac{\varepsilon_{p}^{*}\;-\;\varepsilon_{m}^{*}}{\varepsilon_{p}^{*}{\;+\;2\varepsilon }_{m}^{*}}$$

In Equation (2), ω=2πf is the angular frequency of the applied field (f—frequency of the field); $\varepsilon^{*}=\varepsilon^{\prime }-i\varepsilon^{\prime \prime }$, is the complex dielectric permittivity (subscripts p and m denote the particle and the suspending medium, respectively);$\;i=\sqrt{-1}$ and $\varepsilon^{\prime }$ and $\varepsilon^{\prime \prime }$, are the real and imaginary components of the complex dielectric permittivity [23]. When $\varepsilon_{p}^{*}$ > $\varepsilon_{m}^{*}$, it indicates that the direction of the DEP force is along the direction of the electric field gradient, and the particles move to the region with the strongest electric field intensity, resulting in positive DEP. When $\varepsilon_{p}^{*}$ < $\varepsilon_{m}^{*}$, it indicates that the direction of the DEP force is opposite to the direction of the electric field gradient, and the particles will move to the area where the electric field intensity is weak, resulting in negative DEP.

DEP has been successfully used in biological [24] medical [25], material preparation [26], and environmental treatment fields [27,28,29], such as enrichment, separation, transport, capture and classification of biological particles and cellular molecules. The great potential shown by this method allows us to consider how it can be applied to the removal of algae from eutrophication water bodies. Here, we report an efficient DEP-based method to directly remove *Anabaena*. In addition, we investigated the effects of screen mesh number, voltage, frequency, initial concentration, and flow-rate conditions on the treatment efficiency of *Anabaena* in a home-made DEP device, aiming to provide an efficient and environmentally friendly new approach to the treatment of cyanobacteria in eutrophication water bodies. In this work, the DEP method effectively traps *Anabaena* at the electrode and does not disrupt the biological structure of the cells. For eutrophication water, the cyanotoxins released by cyanobacteria cell rupture have always been an important factor restricting the application of traditional methods.

## 2. Materials and Methods

### 2.1. Materials

The following materials were used in this work: *Anabaena flos-aguas* (Institute of Hydrobiology, Chinese Academy of Sciences, FACHB-245), deionized water (homemade in laboratory), BJ11 medium (excellent purity, Chinese Pharmaceutical Group), acetone (analytical reagent, Chinese Pharmaceutical Group) and 306 stainless-steel wire mesh (30 mesh and 80 mesh, respectively).

### 2.2. Preparation of Solution

*Anabaena* was cultured in BG11 medium, where the light radiation intensity was maintained at 1150–1250 lux. The culture temperature was 25 ± 1 °C in a light–dark cycle (16 h light and 8 h dark). The culture medium and *Anabaena* solution were prepared in different volume ratios to form different concentrations of the initial solution for the DEP experiment.

### 2.3. DEP Experiments

Our DEP experiments were carried out in a home-constructed apparatus, as shown in Figure 1. First, 300 mL of Anabaena solution at a fixed initial concentration is passed into the storage tank at room temperature. Subsequently, a peristaltic pump caused the solution to flow through the DEP vessel at a flow rate of 0.503 L/h, except for the experiment for the effects of flow rate. Ten stainless-steel wire mesh electrodes were installed on the side walls of the DEP vessel, with spacing of 10 mm between adjacent electrodes. The voltages were supplied by an AC power device. Batch experiments were conducted by applying different voltages from 0 V to 15 V and varying the frequency to investigate the effect of DEP force on the *Anabaena* removal effect. Finally, the DEP-treated *Anabaena* solution flowed into the collection pool.

The mesh electrodes were removed from the vessel when the experiment was finished, and the morphology and enrichment of the captured *Anabaena* were observed with a metallographic microscope (ZYJ-330). The content of chlorophyll-a in the sample was determined by spectrophotometry (HJ897-2017). The *Anabaena* solution collected in the collection pool was filtered through a 0.45 μm glass fiber filter membrane. The filtered sample was dried and transferred to a mortar; an amount of 90% acetone solution was added and ground to extract chlorophyll-a. After centrifugation treatment on the extracted solution (adjusted to 10 mL), the absorbance of the extract was determined by UV-Vis spectrophotometer (JASCOV-750) at the wavelengths of 750 nm, 664 nm, 647 nm and 630 nm, respectively. The concentration of chlorophyll-a in the test sample was calculated using the equation below:(3)$$C=11.85{\;\times \;(A}_{664}-A_{750})-1.54{\;\times \;(A}_{647}-{\;A}_{750})-0.08{\;\times \;(A}_{630}-{\;A}_{750})$$
where *C* is the mass concentration of chlorophyll-a in the test sample; *A*_664_, *A*_750_, *A*_647_, *A*_630_ are the absorbance values of the sample at the corresponding wavelengths.

The removal efficiency can be calculated according to the following formula:(4)$$Re=\frac{C_{0}-C_{1}}{C_{0}}\times 100\%$$

Here, *Re* is the Removal efficiency, *C*_0_ represents the mass concentration of chlorophyll-a in the initial sample and *C*_1_ represents the mass concentration of chlorophyll-a in the treated sample.

## 3. Results and Discussion

### 3.1. Direct Capture of Anabaena by DEP

Under the conditions of 15 V AC voltage, frequency of 10 kHz, 0.503 L/h flow rate and 30-mesh electrodes, the experiment was first conducted to verify the possibility of direct capture of *Anabaena* by the DEP. Figure 2a,b show that filamentous and agglomerated *Anabaena* were indeed captured on the wire mesh. Similarly, under the conditions of AC voltage of 15 V, frequency of 10 kHz, flow rate of 0.503 L/h, and 80-mesh electrodes, algal *Anabaena* solution was used to perform the capture test and the results are shown in Figure 2c,d. It is seen that on the stainless-steel wires, directly captured *Anabaena* clusters and single bead-shaped *Anabaena* were clearly presented.

The above experimental results confirm that the direct capture of *Anabaena* by DEP is feasible. Under identical experimental conditions, the aggregated *Anabaena* could be captured by 30-mesh electrodes, while the dispersed *Anabaena* could be captured by 80-mesh electrodes. This result can be understood by considering the formula of the DEP force (Equation (1)). When the wire electrode has a lower mesh count, the pores on the wire are larger in size. The electric field intensity gradient (∇*E_rms_*) produced is relatively smaller at the same voltage. On the other hand, the larger the tested particle’s radius, the greater the exerted DEP force, so the aggregated *Anabaena* are relatively easier for the 30-mesh electrodes to capture compared with the 80-mesh one. In contrast, when the wire electrode has a higher mesh count, the wire network has smaller pores and ∇*E_rms_* is relatively larger. In this case, the dispersed *Anabaena* species, although they have smaller radius and are subject to smaller DEP force, were still captured by the 80-mesh electrodes. Moreover, as can be seen from Figure 2, both aggregated and dispersed *Anabaena* were captured on the cathode and anode, and they were deposited on the surface of the wires rather than stuck in the pores, indicating that a positive DEP process had occurred [30].

In addition, as shown in Figure 3, the *Anabaena* cells captured by the electrodes are intact; they are still spherical or oval in shape. Compared with the *Anabaena* cells without DEP treatment, the cell morphology did not show observable change. This suggests that *Anabaena* cells captured by DEP were highly likely to retain their original cellular structure without any release of cystine arising from the efflux of intracellular substances. Compared with the electrochemical oxidation method [31], this DEP method can effectively avoid the release of algal toxins caused by the rupture of algal cells, thereby effectively reducing the generation of secondary pollution.

### 3.2. Improvement of Dep Device and Its Experimental Results

In previous studies on the removal of heavy metals by DEP, the mesh electrodes with the same size were used to capture the heavy metals [27]. As it is difficult to adequately capture cyanobacterial cells with the same size mesh electrodes, the DEP device was then improved in this work. As shown in Figure 4, 30-mesh wire mesh electrodes were first used to capture aggregated (large size) *Anabaena* in the DEP treatment; filamentous *Anabaena* (small size) were subsequently captured at a later stage using 80-mesh wire mesh electrodes. To compare the filtration effect of 30-mesh, 80-mesh, and 30/80-mesh electrodes alone with the removal effect after applying DEP, we passed *Anabaena* solution with an initial concentration of 756 μg/L into the device at a flow rate of 0.503 L/h. The voltage was fixed at 15 V and the frequency was 10 kHz. The results are shown in Figure 5.

Figure 6a shows a video screenshot of the experiment at 47 s to directly capture *Anabaena* in the experiment using the improved DEP device. It can be seen that when the *Anabaena* solution is pumped into the improved DEP vessel, the larger *Anabaena* cluster was first captured on the mesh electrodes near the inlet. Figure 6b is a screenshot of the 249th second of the same video. Compared with Figure 6a, the number of *Anabaena* captured on the mesh electrodes near the inlet had increased significantly. Compared with the *Anabaena* solution near the inlet of the DEP vessel (the left of the picture), the solution near the outlet (the right of the picture) is much clearer. There were no obvious *Anabaena* organisms that could be observed with naked eyes, which indicates that our experimental method of directly capturing *Anabaena* directly is working.

### 3.3. The Effect of Voltage

Under the condition that the flow rate of suspension was 0.503 L/h, the frequency was 10 kHz, and the initial concentration of Anabaena solution was 756 μg/L, the effect of voltage on the removal of Anabaena was investigated. The results are shown in Figure 7. When the AC voltage applied to the mesh electrodes increased from 0 V to 15 V, the removal rate of chlorophyll-a increased from 68.37% to 89.79%, and the residual concentration of chlorophyll-a decreased from 239 μg/L to 77 μg/L. This shows that the AC voltage in the range of 0–15 V has a significant effect on the removal rate. The higher the voltage, the better the removal rate. This can be explained by the fact that the particles were polarized in solution after application of an external voltage to the solution, and they underwent a DEP-drived migration in the non-uniform electric field. It is indicated by Equation (1) that the magnitude of the DEP force acting on suspended particles is positively correlated with the strength of the non-uniform electric field, which is influenced by the voltage in DEP experiments. The DEP removal rate of *Anabaena* at 15 V is over 20% higher than that obtained via a pure filtration method. Having said that, we considered that high voltages could have a risk of causing corrosion of the electrodes, thus reducing the service life of the electrodes. Moreover, energy consumption and safety are also issues to be considered in practical applications. Therefore, we did not continue to increase the voltage and chose to fix the voltage at 15 V in the subsequent experiments to investigate the effect of other treatment factors on the removal efficiency. It is worth noting that we use an AC power supply that does not exceed 180 mA of current in the 15 V range, which meets the needs of practical applications.

From the viewpoints of both energy consumption and removal rate, we chose 15 V as the best voltage for the direct removal of *Anabaena* by DEP.

### 3.4. The Effect of Frequency

Based on Equations (1) and (2), one can see that the frequency of voltage also directly affects the DEP force, and this frequency effect is multifaceted. Changing the frequency can change the polarization property and the mobility of the particles, and even the direction of DEP [32,33]. We also explored the effect of frequency in this DEP experiment. It can be seen from Figure 8 that for the Anabaena solution with an initial concentration of 1987 μg/L, in the AC voltage range between 3 V and 15 V (0.503 L/h), the removal rate of chlorophyll-a obtained at 100 kHz is higher than that at 10 kHz. This can be explained by the fact that in high-frequency conditions, the effective polarization rate depends mainly on the dielectric constant. If the dielectric constant of the particle is greater than the dielectric constant of the medium, the effective polarization rate will be positive, at which time a positive DEP effect occurs and the cells will move to the region of high electric field strength, and the number of cells captured by the electrodes will increase with frequency [34]. The same conclusion was also drawn by Labeed et al. [35]. Therefore, we chose 100 kHz as the operating frequency in this experiment.

### 3.5. The Effect of Initial Concentration

Using various volume ratios of *Anabaena* solution to the culture medium, we prepared the initial solutions with a chlorophyll-a concentration of 475 μg/L, 949 μg/L, 1424 μg/L, 1899 μg/L and 2373 μg/L. The effect of the initial concentration on the removal rate was investigated at 15 V, 100 kHz and 0.503 L/h. It can be seen from Figure 9 that with the increase in the initial chlorophyll-a concentration, the removal rate of *Anabaena* blooms first increased and then decreased. Within the range of 475–1899 μg/L of chlorophyll-a concentration, this variation trend of the removal rate was positively correlated with the initial concentration, with the removal efficiency increased from 17.05% to 60.95%. However, when the concentration of chlorophyll-a was further increased to 2373 μg/L, the removal efficiency dropped to 55.75%. This may be due to the weaker polarization-sensing effect between adjacent cells at lower concentrations, resulting in fewer cells being polarized, making the DEP effect less pronounced. Additionally, as the concentration increases, the polarization induction effect is enhanced, allowing more cells to undergo dielectric migration. The overall result is that when the initial concentration increased by a 5-fold, i.e., from 475 μg/L to 2373 μg/L, the actual removal efficiency increased by approximately 16-fold, indicating that our DEP method is very suitable for treating *Anabaena* solution in the tested concentration range.

### 3.6. The Effect of Flow Rate

It can be seen from Figure 10 that the removal rate of *Anabaena* was increased from 38.76% to 80.18% when the flow rate of the simulated *Anabaena* solution changed from 0.838 L/h to 0.168 L/h under the test condition of 15 V, 100 kHz, and an initial concentration of 615 μg/L. This shows that the flow rate has a significant effect on the removal rate: the slower the flow rate, the higher the removal rate of *Anabaena*. This can be explained by the fact that as DEP force was the short-distance force, a smaller flow rate would mean a longer processing time, during which *Anabaena* could have more chance of moving to the area of electrodes and then being trapped by the electrodes, and consequently removed from the solution 29. Considering the time factor in an actual treatment, we chose 0.168 L/h as the optimal flow rate for DEP to directly remove *Anabaena*. It is noteworthy that this DEP method shows better removal efficiency compared to the locally enhanced electric field method used by Liu et al. [36] for removing algae from the eutrophication water body.

## 4. Conclusions

In this work, a new method for direct capture and removal of *Anabaena* by DEP was established, and the factors affecting the removal efficiency were investigated. The results showed that the removal rate could be increased from 68.37% without DEP to 89.79% with DEP under identical experimental conditions. Importantly, the increased voltage and frequency in our experimental range enhanced the removal of *Anabaena* by DEP. Moreover, higher initial concentration enhanced the polarization induction effect between adjacent cells, which was reflected in a gradual increase in the actual removal capacity of the DEP method. In addition, if other conditions are fixed, lower flow rates resulted in higher removal rates, as *Anabaena* may have more opportunities to be captured by the electrodes. By adjusting the flow rate, the removal rate of *Anabaena* could increase from 38.76% to 80.18%. Notably, the morphology of the captured *Anabaena* was intact, which showed no sign of the release of algal toxins.

All these suggest that the DEP method has high potential for industrial applications for the treatment of eutrophication water bodies. However, due to practical limitations, it is currently limited to the laboratory. In the future, our focus will be on scaling up experimental equipment to treat eutrophication water bodies and improving electrode materials to improve removal efficiency and save energy.

## Figures and Tables

**Figure 1 ijerph-20-00832-f001:**
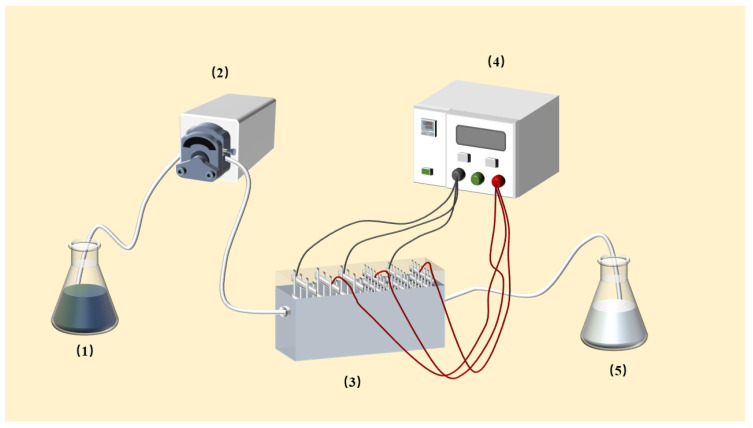
Schematic diagram of the DEP apparatus: (1) storage tank; (2) peristaltic pump; (3) DEP vessel; (4) direct current power source; (5) collection pool.

**Figure 2 ijerph-20-00832-f002:**
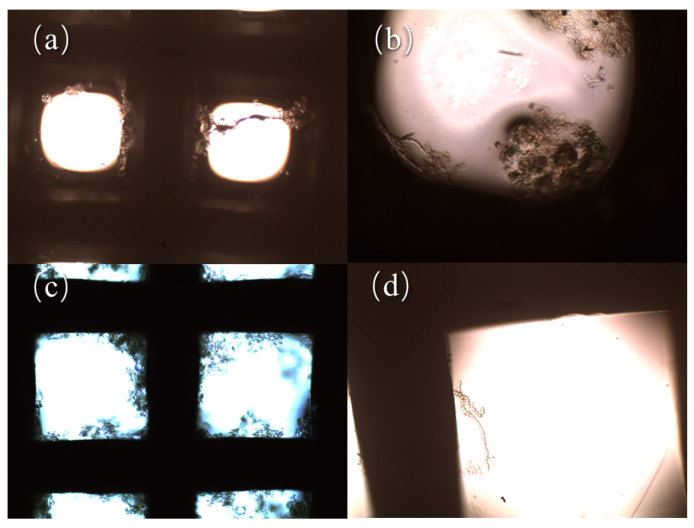
Microscopic images of *Anabaena* captured by the mesh electrodes in the DEP process: (**a**) filamentous *Anabaena* captured by 30-mesh electrodes; (**b**) agglomerated *Anabaena* captured by 30-mesh electrodes; (**c**) *Anabaena* clusters captured by 80-mesh electrodes; (**d**) single bead-shaped *Anabaena* captured by 80-mesh electrodes.

**Figure 3 ijerph-20-00832-f003:**
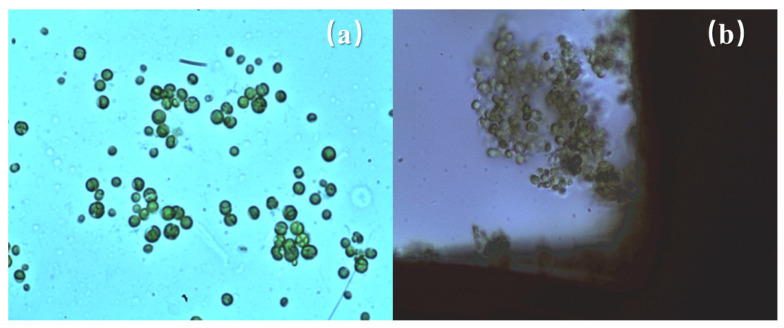
(**a**) Microscopic image of *Anabaena* cells without DEP treatment; (**b**) microscopic image of DEP-treated *Anabaena* cells. (400 times).

**Figure 4 ijerph-20-00832-f004:**
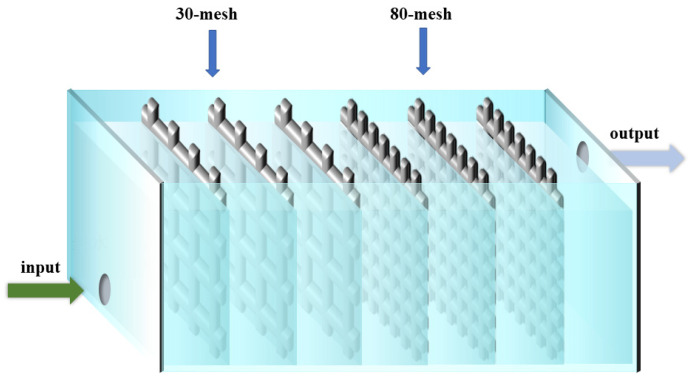
Schematic diagram of the improvement of the treatment pool of the DEP device.

**Figure 5 ijerph-20-00832-f005:**
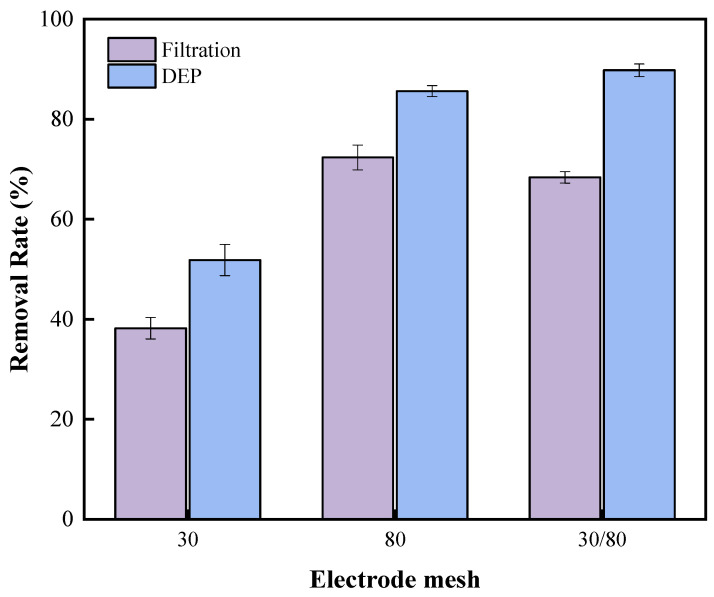
Comparison of the effect of electrodes filtration and DEP with different mesh size.

**Figure 6 ijerph-20-00832-f006:**
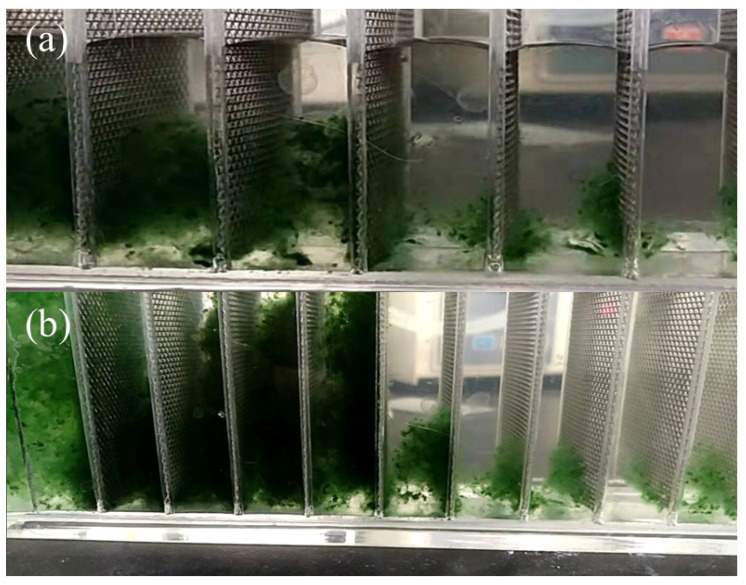
(**a**) Video screenshot of direct capture and removal of *Anabaena* by DEP (47 s); (**b**) video screenshot of direct capture and removal of *Anabaena* by DEP (249 s).

**Figure 7 ijerph-20-00832-f007:**
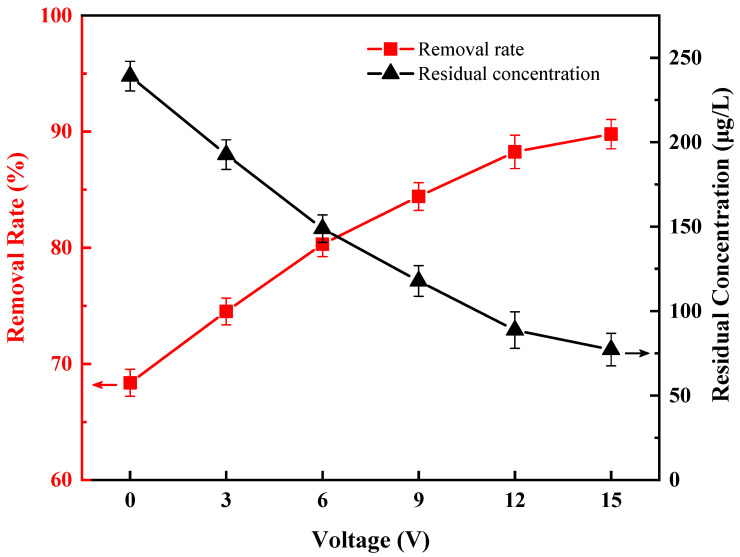
Effect of voltage on direct removal of *Anabaena* by DEP.

**Figure 8 ijerph-20-00832-f008:**
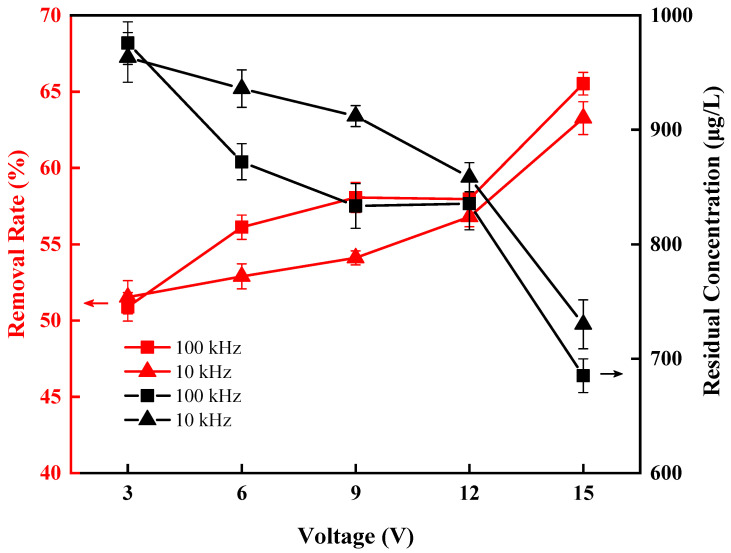
Effect of frequency on direct removal of *Anabaena* by DEP.

**Figure 9 ijerph-20-00832-f009:**
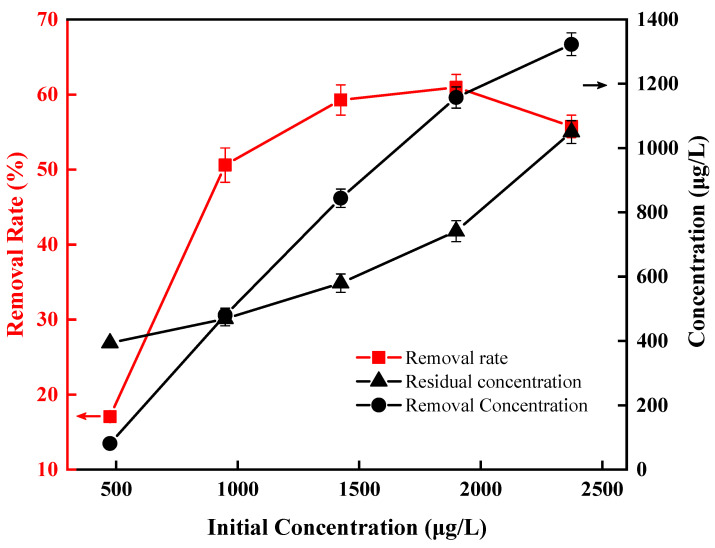
Effect of initial concentration on direct removal of *Anabaena* by DEP.

**Figure 10 ijerph-20-00832-f010:**
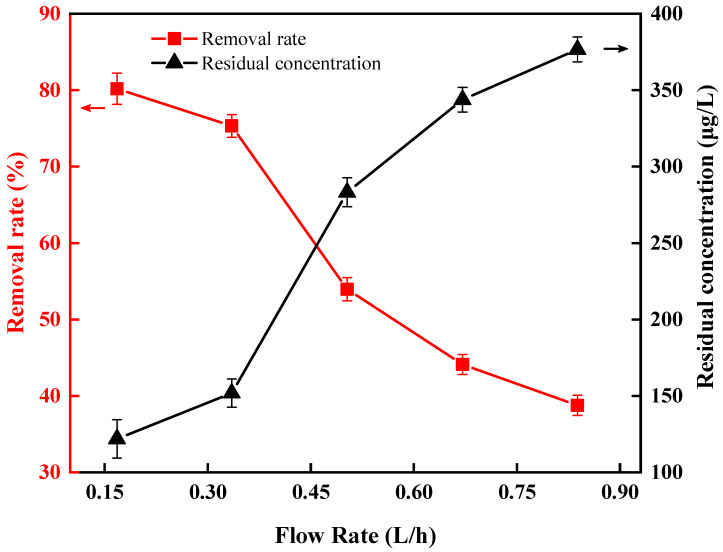
Effect of flow rate on direct removal of *Anabaena* by DEP.

## Data Availability

Not applicable.

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
