# Peer review of "Fast Capture and Efficient Removal of Bloom Algae Based on Improved Dielectrophoresis Process"

_ijerph, 2023, doi:10.3390/ijerph20010832_

Round 1

Reviewer 1 Report

The authors mainly establised dielectrophoresis (DEP) method for direct capture and fast removal of Anabaena. And the factors affecting the removal efficiency of Anabaena were investigated systematically, leading to the optimized experiment conditions and improved DEP process equipment. The topic of this manuscript really fit within the scope of International Journal Environmental Research and Public Health. In addition, further improvement is needed in whole manuscript.

Specific comments are as following:

1、 Introduction:

I think it is necessary to simply state the mechanisms of this technique in this part. And the related literatures need be updated.

2、 Introduction:

It is necessary to make very clear the objectives and need do which work to achieve the previous objectives and finally show the significance of this study.

3、 Materials and methods:

I think this part needs substantial revision. All experiments and methods involved in this paper are not clearly described. Please check it thoroughly.

4、 Results and discussion:

The authors should add more discussion in the explanation of the results, since in the work they found very interesting data that must be discussed in greater depth.

5、 Results and discussion:

This logic of this part is not well, and the connection between the result and the discussion should be well improved.

6、 Conclusions:

This part should be condensed with main findings.

Author Response

We sincerely appreciate your valuable comments and please see the attachment.

Reviewer 2 Report

The author proposed a method for capture and removal of Anabaena with dielectrophoresis (DEP). The effect of electrodes mesh, voltage, frequency, initial concentration and flow rate on the removal efficiency of Anabaena were investigated. The results are interesting. Some problems in the manuscript need to be clarified before it was accepted.

1.     The filtering effect of 30 mesh and 80 mesh electrodes alone needs to be compared with that of DEP.

2.     How long does the experiment last? Will electrochemical corrosion occur to the electrode?

3.     How much is the current? The energy consumption of the process needs to be provided.

4.     P3 line 100 and 101 “306 stainless steel wire mesh (30 mesh, 60 mesh, and 80 mesh, respectively).” There is no data of 60 mesh electrode in the text. “60 mesh” needs to be deleted or supplemented.
